# SketchODE: Learning neural sketch representation in continuous time

**Ayan Das[1,2], Yongxin Yang[1,3], Timothy Hospedales[1,3], Tao Xiang[1,2] & Yi-Zhe Song[1,2]**
[1]SketchX, CVSSP, University of Surrey, UK
[2]iFlyTek-Surrey Joint Research Centre on Artificial Intelligence
[3]School of Informatics, University of Edinburgh, UK
`a.das@surrey.ac.uk, {yongxin.yang, t.hospedales}@ed.ac.uk,`
`{t.xiang, y.song}@surrey.ac.uk`

## Abstract

Learning meaningful representations for chirographic drawing data such as sketches, handwriting, and flowcharts is a gateway for understanding and emulating human creative expression. Despite being inherently continuous-time data, existing works have treated these as discrete-time sequences, disregarding their true nature. In this work, we model such data as continuous-time functions and learn compact representations by virtue of *Neural Ordinary Differential Equations*. To this end, we introduce the first continuous-time Seq2Seq model and demonstrate some remarkable properties that set it apart from traditional discrete-time analogues. We also provide solutions for some practical challenges for such models, including introducing a family of parameterized ODE dynamics & continuous-time data augmentation particularly suitable for the task. Our models are validated on several datasets including VectorMNIST, DiDi and *Quick, Draw!*.

## 1 Introduction

Drawing-based communications such as sketches, writing, and diagrams come naturally to humans and have been used in some form since ancient times. Modeling such data is becoming an increasingly important and topical challenge area for machine learning systems aiming to interpret and simulate human creative expression. Such chirographic structures are challenging to interpret and generate due to their complex and unstructured nature. However recent progress has been strong, thanks to the advances in learning latent representations for sequences (Bowman et al., 2016; Graves, 2013; Srivastava et al., 2015). In particular, with the advent of variational sequence-to-sequence generative models (Srivastava et al., 2015; Bowman et al., 2016) and the collection of large-scale datasets (Gervais et al., 2020; Ha & Eck, 2018; Ge et al., 2021), we have seen a surge of advance-

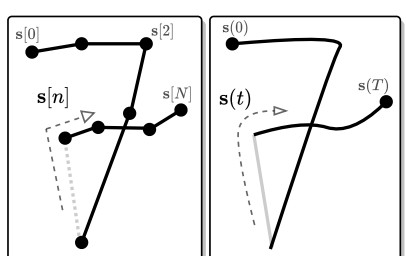

Figure 1: Left: Discrete sequence representation. Right: More natural continuous-time *functional* representation. The gray line denotes pen-up state.

ments (Ha & Eck, 2018; Aksan et al., 2020) in this direction. Nevertheless, a key missing link is the fact that drawing is intrinsically continuous in time, as is the resulting drawn artefact, when considered as a sequence rather than a raster image.

The dominant approach to model free-hand sketches, popularized by Ha & Eck (2018), has been to use a discrete-step recurrent network for capturing the latent process $p(\mathbf{h}[n]|\mathbf{h}[n-1])$ with discrete step $n$, and an observation model $p(\mathbf{s}[n] \mid \mathbf{h}[n])$ to explain local structure. The large body of methods built upon this have a critical flaw: they ignore the very nature of the data, i.e. drawn structures are inherently *continuous* in time (Refer to Fig. 1). While a few attempts have been made to use continuous-time constructs like Bézier curves (Das et al., 2020; 2021), their inconvenient mathematical form affect the representational power as well as training dynamics. The reason for such an absence of continuous-time models is the lack of fundamental tools for handling continuous time

data. Lately however, the introduction of *Neural Ordinary Differential Equations* (Chen et al., 2018) followed by numerous extensions (Dupont et al., 2019; Yildiz et al., 2019; Kidger et al., 2020) have opened the possibility of building powerful models that can *natively* represent continuous time data.

In this paper, we represent chirographic drawing data like handwriting, sketches etc. as continuous-time function $\mathbf{s}(t)$ and model the underlying latent process also as continuous-time latent function $\mathbf{h}(t)$. Specifically, we propose a framework for capturing the derivative (potentially higher order) of the latent process $\frac{d^K \mathbf{h}(t)}{dt^K}$ with parameterized neural networks. At inference, a solution trajectory $\hat{\mathbf{s}}(t)$ is computed from a given Initial Value Problem (IVP) that includes a learned ODE dynamics and an initial hidden state $\mathbf{h}_0 \triangleq \mathbf{h}(t = 0)$. An obvious advantage of the functional representation is its arbitrary spatial resolution, i.e. we can retrieve a structure at any spatial resolution by sampling the function at appropriate rate. Moreover, with a functional representation, we can systematically upgrade the representation along time with additional properties (e.g. line thickness, color etc).

So far, parameterized ODE dynamics models have largely been treated as a "replacement for ResNets" (Massaroli et al., 2020) where the intermediate states are of little importance. While the capability of ODE models are beginning to be tested in time-series data (Kidger et al., 2020; Morrill et al., 2021), they still remain largely unexplored. Following the introduction of Neural ODEs by Chen et al. (2018), methods have been proposed in order to regularize the solution trajectory to be *as simple as possible* (Kelly et al., 2020; Finlay et al., 2020). However, time-series data such as chirography are entirely the opposite. The dynamics model, and consequently the solution trajectory should be *as flexible as possible* in order to learn the high degree of local and global variations with time often exhibited in such data. We increase the flexibility of Neural ODE models by introducing a new class of parameterized dynamics functions. Our experiments show that this is crucial in order to model chirographic data with satisfactory fidelity. We also introduce a new data augmentation method particularly geared towards ODE models and continuous-time sequences.

Finally, we propose a deterministic autoencoder with a global latent variable $\mathbf{z}$ that exhibits some generative model properties by virtue of inherent continuity. Our final model, SketchODE, is similar to SketchRNN (Ha & Eck, 2018) in terms of high-level design but lifts it to the realm of continuous-time. Even though employed for chirographic data in this paper, our core model is the first generic continuous time Seq2Seq architecture. We explore some of the noteworthy features that differentiate SketchODE from discrete-time Seq2Seq analogoues.

**An inductive bias for continuity:** Discrete-time sequence models (Ha & Eck, 2018) need to use their capacity to model both *global structure* and *temporal continuity* in the data. ODE models on the other hand, have a strong intrinsic architectural bias towards temporal continuity. Sequences generated using a well-behaved ODE dynamics function are guaranteed to be continuous. Since they need not learn the continuity bias from scratch, ODE models are more data-efficient and are able to use majority of their capacity for modelling higher level structures. We even demonstrate that our SketchODE supports meaningful 1-shot learning when fine-tuned.

**Deterministic Autoencoding with Structured Latent Space:** A surprising property of Seq2Seq ODE models is the ability to perform latent space interpolation and sampling *without* the necessity of imposing a prior distribution $p(\mathbf{z})$ on the variational posterior $q_\phi(\mathbf{z}|\mathbf{s})$. This property is also a consequence of the latent to output mapping being continuous.

## 2 RELATED WORK

With the advent of recurrent neural networks, sequential time-series models have seen significant adaptation in a variety of applications. Video (Srivastava et al., 2015) and natural language (Bowman et al., 2016) were two early applications to have seen significant success. A different but important time-series modality is chirographic data like handwriting (Graves, 2013) and sketches (Ha & Eck, 2018) which gained traction only recently. Subsequent developments also address data modalities that are *not* free-flowing, like Fonts (Lopes et al., 2019) and Icons (Carlier et al., 2020) which require slightly different models. Mirroring developments in natural language processing (Vaswani et al., 2017), Transformers are now replacing (Carlier et al., 2020) recurrent networks in such models.

Following the seminal work of Ha & Eck (2018), the primary representation for chirographic data has been discrete sequence of waypoints or tokens from specialized Domain Specific Languages like SVG (Scalable Vector Graphics), with a minority continuing to make use of standard raster graphic

architectures (Ge et al., 2021). Overall, few studies have attempted to develop better representations using specialized functional forms, with Aksan et al. (2020) learning stroke embeddings and Das et al. (2020; 2021) directly modeling strokes as parametric Bézier curves. However, to the best of our knowledge, modelling chirographic data as generic functions have not yet been tried so far.

The Neural ODE (Chen et al., 2018) provides a credible tool for natively modelling continuous-time functions using dynamical systems. Since their inception, Neural ODEs have sparked a wide range of discussions and developments (Massaroli et al., 2020), mostly in the theoretical domain. Neural ODEs have been extended to work with latent states (Dupont et al., 2019), to define generative models (Song et al., 2021), and have external data control (Kidger et al., 2020). While their practical training remain a challenge, significant amounts of work have been dedicated towards developing engineering techniques (Finlay et al., 2020; Grathwohl et al., 2019; Kelly et al., 2020; Poli et al., 2020) for faster convergence and better regularization. Despite flourishing theoretical interests, due to high training complexity, applied works have so far been limited to few domains like Reinforcement Learning (Du et al., 2020), reasoning in visual question answering (Kim & Lee, 2019), and physical system identification (Lutter et al., 2019; Greydanus et al., 2019; Finzi et al., 2020) etc.

## 3    NEURAL ORDINARY DIFFERENTIAL EQUATIONS (NEURAL ODE)

Neural ODE (Chen et al., 2018) provides a framework for modelling an inherently continuous-time process by means of its time derivative. Given a continuous-time process $\mathbf{s}(t) \in \mathbb{R}^d$ over a defined time interval $[t_0, t_1]$, we can approximate its underlying true dynamics by learning the parameters of a neural network $\mathcal{F}_\Theta(\cdot)$ acting as a proxy for the same

$$\dot{\mathbf{s}}(t) = \mathcal{F}_\Theta(t, \mathbf{s}) \in \mathbb{R}^d. \tag{1}$$

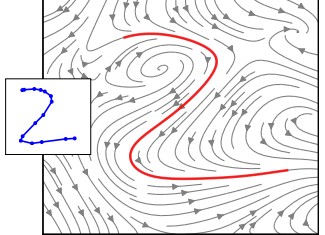

The time derivative of a vector-valued function, i.e. $\dot{\mathbf{s}}(t)$ defines a *vector field* over $\mathbb{R}^d$ which guides the evolution of the process over time. Given the initial state $\mathbf{s}_0 := \mathbf{s}(t = t_0)$ and the learned set of parameters $\Theta^*$, the process can be recovered by solving an Initial Value Problem (IVP) using any Differential Equation solver

$$\mathbf{s}(t) = \mathbf{s}_0 + \int_{t_0}^{t} \mathcal{F}_{\Theta^*}(t, \mathbf{s}) \cdot dt \ \ \forall t \in [t_0, t_1]$$

Figure 2: Vector-field depicting dynamics directly on $\mathbf{s}(t)$; modelled trajectory (red) learned from VectorMNIST data (blue).

Given a set of time-varying functions, one can learn dynamics $\mathcal{F}_\Theta$ capturing high-level concepts. The learning algorithm provided by Chen et al. (2018) makes it practical to train such models with constant memory cost (w.r.t time horizon) both in forward and backward pass.

## 4    FRAMEWORK OVERVIEW

We borrow the notation of Section 3 and denote chirographic structures as continuous functions of time $\mathbf{s}(t)$ defined within $[t_0, t_1]$. The exact form of $\mathbf{s}(t)$ must include the running $2D$ coordinates over time and may include other properties like pen-state (whether ink is visible at time $t$) etc. In principle, one could directly model $\dot{\mathbf{s}}(t)$ with a Neural ODE as in Eq. 1 (see Fig. 2). However, such a model would possess low representational power resulting in underfitting, and furthermore would be incapable of representing self-overlaps naturally exhibited by chirographic data.

### 4.1    NEURAL ODE AS DECODER

**Augmented ODE.**   We first *augment* (Dupont et al., 2019) the original state $\mathbf{s}(t)$ with the underlying hidden process $\mathbf{h}(t)$ in the same time interval. Unlike discrete-time analogues like recurrent networks which model discrete transition of the hidden state followed by an observation model $p(\mathbf{s}|\mathbf{h})$, we model the "co-evolution" of the hidden process along with data using a dynamical system defined over the joint space $\mathbf{a}(t) := [\mathbf{s}(t) \ \mathbf{h}(t)]^T$. Augmented ODEs (Dupont et al., 2019) improve representational power by lifting the vanilla ODE of Eq. 1 to a higher dimension, allowing the data trajectory to *self-overlap* in time. In addition, an augmented latent state $\mathbf{h}(t)$ provides a principled way to *control* the trajectory by simply solving the dynamics with a different initial condition $\mathbf{h}_0$.

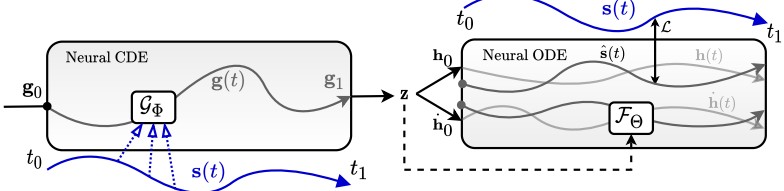

Figure 3: Overview of the SketchODE framework. The Neural CDE encodes input sequence $\mathbf{s}(t)$ to a latent vector $\mathbf{z}$ and a second-order Augmented Neural ODE decodes it as $\hat{\mathbf{s}}(t)$.

**Higher-Order Dynamics.** To further increase representational power, we capture the second-order derivative $\ddot{\mathbf{a}}(t)$ with a parameterized neural network $\mathcal{F}_\Theta(\cdot)$ like Yildiz et al. (2019). This formulation however, leads to the requirement of an extra initial condition $\dot{\mathbf{a}}_0 := \dot{\mathbf{a}}(t = t_0)$

$$\begin{bmatrix} \mathbf{a}(t) \\ \dot{\mathbf{a}}(t) \end{bmatrix} = \begin{bmatrix} \mathbf{a}_0 \\ \dot{\mathbf{a}}_0 \end{bmatrix} + \int_{t_0}^{t} \begin{bmatrix} \dot{\mathbf{a}}(t) \\ \mathcal{F}_\Theta(\cdot) \end{bmatrix} dt \tag{2}$$

Traditionally, the quantities $\mathbf{a}(t)$ and $\dot{\mathbf{a}}(t)$ are termed as "position" and "velocity" respectively of the dynamical system represented by $\ddot{\mathbf{a}}(t) = \mathcal{F}_\Theta(\cdot)$. Inspired by SketchRNN (Ha & Eck, 2018), we simplify the model by dropping the position component $\mathbf{a}(t)$ as a dependency to the dynamics function, leading to a dynamics of $\ddot{\mathbf{a}}(t) = \mathcal{F}_\Theta(\dot{\mathbf{a}}(t), t)$.

**Conditioning.** Our model so far is deterministic given a particular choice of initial condition. In order to generate different samples, we need to condition the decoding on a latent vector. Thus, we introduce the following architectural changes:

1. We define a *global* latent vector $\mathbf{z}$ for each sample and compute the hidden part of initial conditions by projecting it through $\mathbf{L}_{dec}$, as $\mathrm{H}_0 := [\mathbf{h}_0 \ \dot{\mathbf{h}}_0]^T = \mathbf{L}_{dec}(\mathbf{z})$. Without loss of generality, we fix the visible part of the initial state, i.e. $\mathbf{s}_0$ and $\dot{\mathbf{s}}_0$ to be constants.

2. We further include $\mathbf{z}$ directly as a fixed parameter to the dynamics function. This is a specific form of *data-controlled* dynamics (Massaroli et al., 2020). Please see Appendix B for implementation details.

The final second order dynamical system represented by our decoder is therefore

$$\ddot{\mathbf{a}}(t) = \mathcal{F}_\Theta(\dot{\mathbf{a}}(t), t, \mathbf{z}) \tag{3}$$

Computing the forward pass (Eq. 2) requires the knowledge of the exact value of $\mathbf{z}$. The latent code can be computed by employing a parameterized encoder. Similar to discrete-time analogues, we propose to employ a non-causal model as an encoder.

## 4.2 NEURAL CDE AS ENCODER

We propose to use *Neural Controlled Differential Equations* or Neural CDEs (Kidger et al., 2020) to map a given data sample $\mathbf{s}^{(i)}(t)$ to its latent code $\mathbf{z}$.

Neural CDEs are dynamical systems where the hidden state $\mathbf{g}(t)$ evolves under the control of a time-varying process. Given a real data sample from the dataset $D = \{\mathbf{s}^{(i)}(t)\}_{i=1}^N$, an initial state $\mathbf{g}_0 := \mathbf{g}(t = t_0)$ is transformed by running an ODE solver on a surrogate ODE dynamics defined as

$$\dot{\mathbf{g}}(t) = \mathcal{G}_\Phi(t, \mathbf{g}) \, \dot{\mathbf{s}}^{(i)}(t)$$

where $\mathcal{G}_\Phi$ is a parameterized neural network that is analogues to a Bidirectional-RNN encoder in discrete-time. The latent code for the data sequence $\mathbf{s}^{(i)}(t)$ is a mapping $\mathbf{L}_{enc}$ of the state that fully encodes the incoming data, i.e. $\mathbf{g}_1 := \mathbf{g}(t = t_1)$, which is given by

$$\mathbf{z} = \mathbf{L}_{enc}(\mathbf{g}_1), \text{ where } \mathbf{g}_1 = \mathbf{g}_0 + \int_{t_0}^{t_1} \mathcal{G}_\Phi(t, \mathbf{g}) \, \dot{\mathbf{s}}^{(i)}(t), \tag{4}$$

where $\mathcal{G}_\Phi$ being a function of time facilitates the usage of non-uniformly sampled data at inference. As suggested by Kidger et al. (2020), the velocities of data trajectory $\dot{\mathbf{s}}^{(i)}(t)$ can be computed with

any non-causal interpolation (e.g. Natural Spline curve) of the original discrete data. Alternatively, with some compromise to accuracy, crude estimates can also be computed from discrete data using forward/backward/central difference operator. Refer to Fig. 3 for an overview of the architecture.

## 4.3 LOSS FUNCTION & TRAINING

Given a ground truth trajectory $\mathbf{s}^{(i)}(t)$, and the reconstructed trajectory $\hat{\mathbf{a}}^{(i)}(t)$ (or implied $\hat{\mathbf{s}}^{(i)}(t)$) computed by executing the encoder and decoder above, we can simply minimize a regression loss (w.r.t $\Theta, \Phi, \mathbf{L}_{enc}, \mathbf{L}_{dec}$) at any granularity depending on availability of data and quality requirement

$$\mathcal{L}_{\text{fit}}\big(\hat{\mathbf{a}}^{(i)}(t), \mathbf{s}^{(i)}(t)\big) = \int_{t_0}^{t_1} \mathcal{L}\big(\hat{\mathbf{s}}^{(i)}(t), \mathbf{s}^{(i)}(t)\big)dt \approx \sum_{t \in [t_0, t_1]} \mathcal{L}\big(\hat{\mathbf{s}}^{(i)}(t), \mathbf{s}^{(i)}(t)\big) \tag{5}$$

**Discussion.** Note that the solution trajectory computed by Eq. 2 is entirely deterministic given the latent code. However, the end-to-end continuity of the formulation allows discovering structured latent space by default. The mapping from the input $\mathbf{s}^{(i)}$ to latent code $\mathbf{z}$, and $\mathbf{z}$ to predicted trajectory $\hat{\mathbf{s}}^{(i)}(t)$ is continuous and smooth as long as the dynamics functions governing the encoder CDE and decoder ODE are lipschitz continuous. Consequently, an infinitesimally small perturbation in latent space leads to an infinitesimally small change in the predicted trajectory. This property is important because it enforces structure in the latent space, enables latent space interpolations, and can be exploited to sample new data – as we will show in Section 5.

Our model can also be instantiated as a stochastic generative model (Kingma & Welling, 2014; Ha & Eck, 2018) by assuming a standard isotropic Gaussian prior on the latent code and parameters of the dynamics. Please refer to Appendix A for the exact factorization of the joint distribution and the resulting ELBO objective $\mathcal{L}_{\text{prob}}$. However, we focus on the deterministic variant in the main paper in order to investigate the unique properties it exhibits, as mentioned in Section 1. This also has the benefit of leading to relatively simple learning dynamics due to the absence of VAE-style KL-divergence loss (Kingma & Welling, 2014), which is known to be notoriously challenging to optimize and require specialized engineering tricks (Higgins et al., 2017; Bowman et al., 2016).

## 4.4 DATA FORMAT

Our model does not make any assumption on the exact form of $\mathbf{s}(t)$. However, we propose two practical data format which might be useful for different applications and lead to different trade-offs between computational cost and representation power.

**Full-sequence format:** $\mathbf{s}(t)$ is comprised of two components, the position of the pen $\mathbf{x}(t)$ and the state of the pen $\mathbf{y}(t) \in \{0, 1\}$ which denotes whether the ink should be rendered or not. Following Section 4.1, we fix $\mathbf{x}_0 = \mathbf{0}$ and $\mathbf{y}_0 = 0$. Please note that $\mathbf{y}(t)$ is different to the traditional "pen-change" state popularized by Ha & Eck (2018) which denotes whether the pen *switched* its state.

**Multi-stroke format:** Alternatively, we may represent each sample as a set of its constituent strokes $\{\mathbf{x}^1(t), \mathbf{x}^2(t), \cdots\}$, where $\mathbf{x} := [x\ y]^T \in \mathbb{R}^2$, each of which is represented as continuous time functions. In order to incorporate such data format, a slightly modified version of the model is required. For CDE based encoder, we encode every stroke $\mathbf{x}^k(t)$ individually using Eq. 4 with initial state set to an updated version of the last state from previous stroke with a simple transformation. A similar approach is followed for the decoder ODE with the end of sample decided using a binary flag predicted from last state of a stroke. Please refer to Appendix C for detailed description.

## 4.5 A FLEXIBLE FAMILY OF ODE DYNAMICS

A vital component of the model with regard to capturing chirographic data properly is the family of parameterized dynamics $\mathcal{F}_{\Theta}$ and $\mathcal{G}_{\Phi}$. So far, the dynamics functions have been chosen to be MLPs (Chen et al., 2018; Massaroli et al., 2020) with standard activation functions like $\text{ReLU}(\cdot)$, $\text{Tanh}(\cdot)$ and $\text{Sigmoid}(\cdot)$, even for time-series data (Kidger et al., 2020; Yildiz et al., 2019). This class of dynamics functions are restricted in terms of the complexity of trajectories they can model and not flexible enough to capture high frequency temporal changes (e.g. multiple overlaps in short time, sharp edges etc.) in the data trajectories. Inspired by SIREN (Sitzmann et al., 2020), which intro-

Figure 4: Left: $2D$ vector fields induced by MLPs (with random weights) with different activation functions. A few randomly sampled solution trajectories are shown below the corresponding vector fields. Right: Original data and augmented versions with different strengths of Perlin Noise.

duced a way to increase spatial frequency content, we propose to use periodic activation functions in order to increase the temporal frequency content of solution trajectories. We use sinusoidal functions, i.e. $\textsc{Sin}(\omega \cdot x)$ or $\textsc{Cos}(\omega \cdot x)$ as pointwise non-linearities. $\omega$ is a parameter of the activation function and controls the extent of flexibility of the dynamics function. $\omega$ can be treated as hyperparameter or can be learned individually for each layer. Fig. 4 (left) visualizes the dynamics functions induced by MLPs with different activation functions as well as the effect of $\omega$ on periodic activation. By controlling $\omega$, it is possible to reduce the frequency content of the solution trajectories under the model, leading to an "abstraction effect" (see Sec. 5.5).

## 4.6 Continuous data augmentation

Data augmentation is performed for increasing the effective number of samples by creating synthetic ones from real data. It aims to create a data cloud around the real samples, helping the model discover local manifolds. Discrete time chirographic models are trained primarily by augmenting sequential data with independent Gaussian noise on its constituent points (Ha & Eck, 2018). We propose a new data augmentation strategy specifically fit for continuous time sequences. We use *Perlin Noise* (Perlin, 1985; 2002), a popular continuous noise algorithm heavily used in Procedural Generation (Lagae et al., 2010), to sample continuous function $\mathbf{p}(t) \in \mathbb{R}^2$ in the same time interval as a given $\mathbf{s}(t)$. The noise function is inherently continuous but can be sampled at any necessary granularity. The noise function is then centered (by subtracting its center of gravity) and added to coordinates of the original sample. Refer to Fig. 4 (right) for visual examples. The simplest way to sample an augmented version is by adding $\sigma \cdot (\mathbf{p}(t) - \mathbf{p}_{CG})$ to the constituent strokes of $\mathbf{s}(t)$, where

$$\mathbf{p}(t) = \begin{bmatrix} \text{Perlin2D}(\mathbf{s}(t) + \epsilon_x + \epsilon) \\ \text{Perlin2D}(\mathbf{s}(t) + \epsilon_y + \epsilon) \end{bmatrix} \text{ and } \mathbf{p}_{CG} = \int_{t_0}^{t_1} \mathbf{p}(t)dt$$

The $\text{Perlin2D}(\cdot)$ is the original *2D Perlin Noise* function in two dimension, $[\epsilon_x, \epsilon_y]$ and $\epsilon$ are coordinate and sample level seeds to simulate pseudo-random behavior. $\sigma$ is the strength of the noise.

## 5 Observations & Experiments

### 5.1 Datasets & Setup

We experimented with three chirographic datasets to evaluate our model and compare it to traditional discrete time models. We use a $80\% - 20\%$ train-test split for each.

**VectorMNIST:** We collected a small scale dataset of vectorized MNIST digits, i.e. handwritten 0 to 9. We recorded dense points along the trajectory to capture its functional form with least degree of error. A total of 1000 instances of MNIST digits were collected. The dataset is then expanded synthetically by augmenting the real data the with Perlin Noise, as explained in Section 4.6.

**QuickDraw:**[1] Released by Ha & Eck (2018), *Quick, Draw!* is the largest collection of publicly available free-hand doodling dataset. These doodles are casual representations of familiar concepts drawn within a short amount of time, and are diverse and free-flowing in nature. We use a few categories (face, fish, flower) in order to compare our model against traditional discrete time alternatives.

---

[1] https://github.com/googlecreativelab/quickdraw-dataset

**DiDi Dataset**[2]**:** Released by Gervais et al. (2020), *Digital Ink Diagram dataset* is a large collection of synthetically generated flowcharts comprising of different components connected by arrows and lines. It has previously been used for modelling compositional relations by Aksan et al. (2020). We use the "unlabeled" version of DiDi dataset which doesn't have labels drawn on the diagrams.

**Implementation Details** To train our framework in practice, we require discretized but possibly non-uniformly sampled ground-truth data along time. For the full-sequence format, each sample in the dataset is represented by an $N$ length sequence each including pen-states. For multi-stroke, each data sample may consists of multiple strokes each represented as a discrete coordinate sequence. For computational and representational convenience, we re-scale the time-range for both full-sequence and every stroke in multi-stroke format to $[0, T]$ with $T$ being a hyperparameter. For time-sensitive applications, time can be left as it is in the raw data and can be regressed from the latent vector (full-sequence format) or initial decoder state (multi-stroke format). Please refer to Appendix D for a full description of the data structures, resampling and batching mechanism used in practice.

Computing the latent code using CDE based encoder requires a functional form $\mathbf{s}(t)$ at training and inference. We use Natural Spline as suggested by Kidger et al. (2020) to construct a spline function from discrete data in order to compute the time-derivative in Eq. 4. For both encoder and decoder, the forward passes involve computing the ODE solution using a black-box ODE solver. Instead of a more accurate *adaptive solver*, we use RK4 (The classic Runge-Kutta method) as it strikes a perfect balance between computation efficiency and solution accuracy.

The pointwise regression loss $\mathcal{L}$ in Eq. 5 is chosen to be Huber Loss or Smooth L1 loss (Girshick, 2015) and the summation is computed for a fixed sequence of uniform time points with granularity $G_{seq}$ for full-sequence format and $G_{stroke}$ for multi-stroke format. The granularity does not affect memory consumption as both forward and backward pass of Neural ODE models are constant-memory wrt to the number of points to be evaluated. Due to complex nature of data, *Quick, Draw!* and DiDi are trained using multi-stroke model with $G_{stroke} = 30$; otherwise $G_{seq} = 75, G_{stroke} = 25$. The dimensionality of the hidden state in CDE/ODE is denoted $d$ and the latent dimension as $l$. For experimentation, we choose $d = 64, l = 48$ for VectorMNIST, $d = 96, l = 84$ for DiDi and $d = 144, l = 96$ for *Quick, Draw!*. Dynamics networks (for both CDE and ODE) are MLPs with few (2 for full-sequence, 1 for multi-stroke) hidden layers of size $\lceil 1.2d \rceil$. For a fair comparison with discrete time models, we use an LSTM-LSTM Seq2Seq model with same hidden dimension. We empirically found the best value of the upper limit of time range to be $T = 5$. All models are trained with AdamW (Loshchilov & Hutter, 2019) optimizer, learning rate of $1 \times 10^{-4}$ for RNN and $3 \times 10^{-3}$ for CDE-ODE, annealed using cosine scheduler (100 epoch period) with decreasing (by factor of 4) amplitude. We also used $10^{-2}$ as weight decay (L2 penalty) regularizer on the weight of the dynamics networks and gradients are clipped at a magnitude of 0.01.

## 5.2 RECONSTRUCTION & GENERATION TASKS

We first evaluate our framework for the traditional reconstruction task and compare with discrete time RNN model. Reconstruction performance is an indicator of a model's ability to faithfully capture an input by means of a compact latent code. We compare our deterministic SketchODE model with an RNN-RNN Seq2Seq model trained for deterministic reconstruction. We also train a recognition CNN on the train data and apply it to the rasterized reconstruction of test data as a semantic measure of reconstruction fidelity. We compute test loss and

|  | Quick,Draw! | | VectorMNIST | | DiDi | |
|---|---|---|---|---|---|---|
|  | Loss | % acc. | Loss | % acc. | Loss | % acc. |
| SketchODE | 0.0056 | 93.89 | 0.0016 | 98.41 | 0.0024 | 96.22 |
| RNN-RNN | 0.0048 | 94.20 | 0.0011 | 98.48 | 0.0020 | 96.24 |
| CoSE | 0.0034 | 95.27 | 0.0007 | 99.02 | 0.0018 | 96.31 |
| BézierSketch | 0.0121 | 82.12 | 0.0034 | 96.04 | 0.0094 | 91.46 |

Table 1: Autoencoding loss and recognition accuracy of reconstructed input between SketchODE and traditional models.

recognition accuracy for both model on all datasets. The results (in Table. 1) show that SketchODE performs comparably to standard RNN-RNN Seq2Seq models. While reconstruction loss is slightly worse, SketchODE retains recognizable visual semantics. This confirms that continuous time models have a strong inductive bias towards semantic content rather than exact point-wise reconstruction. We also compare with two related alternatives: BézierSketch (Das et al., 2020) and CoSE (Aksan et al., 2020) trained deterministically.

---

[2]https://github.com/google-research/google-research/tree/master/didi_dataset

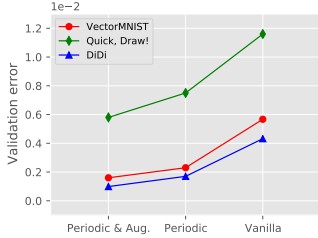 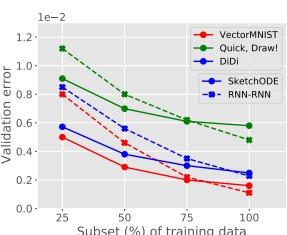

Figure 5: Left: Conditional sampling in SketchODE vs discrete RNN-RNN vs CoSE. Middle: Ablation study to validate the use of periodic activations & Perlin noise augmentation in SketchODE. Right: SketchODE provides more data efficient learning than an RNN-RNN analogue.

We also validated our design choices of periodic activations and continuous noise augmentation for reconstruction task with an ablation study shown in Figure. 5 (middle).

The generative model trained with an extra $\mathcal{L}_{\mathrm{prob}}$ (as described in Section 4.3) as loss can generate plausible data even unconditionally (Refer to Appendix A). However, a special property of continuous time Seq2Seq models like ours is conditional sampling from a *deterministically* trained network. Since the mapping from latent code to output is defined by an ODE solution trajectory, infinitesimal perturbation of latent codes leads to similar perturbation in the data, thus preserving semantic structure. For the discrete counterpart i.e. RNN Seq2Seq models, the sampling-based autoregressive nature of inference makes the latent to output map discontinuous. For CoSE (Aksan et al., 2020), even if the the individual strokes remain continuous, their geometry and relational structures (autoregressive) break down. We deterministically encode input data into latent code $\mathbf{z}$ and sample from $\mathcal{N}(\mathbf{z}, \sigma\mathbf{I})$. Figure 5 (left) shows qualitative results for conditional sampling with $\sigma = 0.05$. Clearly, the autoregressive models (i.e. RNN and CoSE) failed to retain both smoothness and semantics, whereas samples from SketchODE are diverse, visually pleasing and meaningful.

## 5.3 DATA EFFICIENCY

Temporally continuous models are data efficient for chirographic representation due to their strong inductive bias towards continuous data. The implicit distribution defined by our SketchODE model has a support only over all possible *continuous* trajectories. Discrete time sequence models on the other hand, define a distribution over all possible trajectories and hence are required to "learn" the bias (over continuous trajectories) from data itself. To illustrate this we train both RNN-RNN and our SketchODE model on varying amounts of data. Dimensionality of state/hidden vectors are kept to be same for fair comparison. Fig. 5 (right) shows the test reconstruction error as a function of training data. We can see that SketchODE has a particular advantage in the low data regime.

## 5.4 INTERPOLATION & ANIMATION

Continuing the discussion about the strong inductive bias toward continuous data, SketchODE cannot generate temporal discontinuity for any latent vector $\mathbf{z}$. This leads to the property that smooth interpolation in latent codes leads to smooth interpolation in the data. This is a noteworthy properly for a deterministic autoencoder like SketchODE as it is not true for traditional RNN-RNN models. To illustrate this we train SketchODE and extract latent codes $\mathbf{z}_0$ and $\mathbf{z}_1$ for two data samples $\mathbf{s}^{(1)}$ and $\mathbf{s}^{(2)}$. We interpolate between latent codes and produce $\mathbf{z}_{\mathrm{interp}} = \mathbf{z}_0 * \alpha + \mathbf{z}_1 * (1 - \alpha)$ and decode $\mathbf{z}_{\mathrm{interp}}$ for several values of $\alpha \in [0, 1]$. Fig. 6 (left, right) illustrate interpolation in VectorMNIST, QuickDraw and DiDi datasets. More examples are shown in Appendix E.

**One-shot interpolation between unseen classes** Existing latent models like (Ha & Eck, 2018) need to be trained on substantial amounts of data in order to learn the manifold required for latent-space exploration. A unique capability of SketchODE is the ability to perform interpolation after 1-shot learning of new categories. This is due to the strong inductive bias of continuity that must be learned from scratch in discrete-time analogues. We demonstrate this by creating two extra handwritten samples "A" and "B", and fine-tune our trained VectorMNIST model for approx $1k$ iterations. We are then able to interpolate between novel letters as shown in Fig. 6(middle).

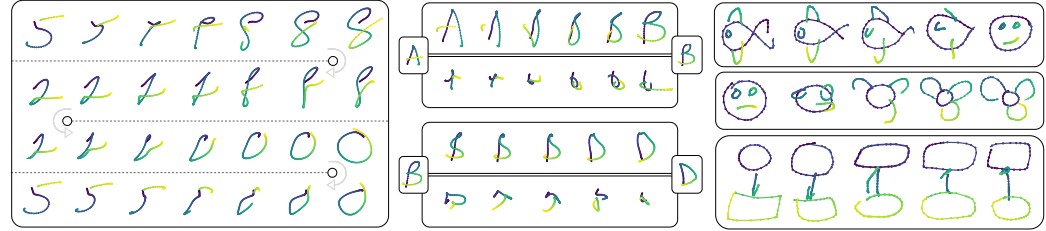

Figure 6: Illustration of SketchODE latent space interpolation and animation. Left: Interpolation $5 \rightarrow 8 \rightarrow 2 \rightarrow 0 \rightarrow 5$ in VectorMNIST. Middle: SketchODE(top) supports interpolation between novel categories (e.g., $A \rightarrow B$ and $B \rightarrow D$) unseen during training (on VectorMNIST), SketchRNN(bottom) fails to do this. Right: Interpolation examples in QuickDraw and DiDi datasets.

## 5.5 ABSTRACTION EFFECT

Strong inductive bias towards temporal continuity enables another capability which we term the "abstraction effect". In the context of sketch, "abstraction" (Berger et al., 2013; Muhammad et al., 2019) refers to removing details from the data while keeping as much semantic content as possible. Our proposed continuous SketchODE model is a perfect fit for creating such abstraction. As discussed before, SketchODE can use almost the entirety of its capacity for modelling the semantic content since temporal continuity comes for free. We observed that by reducing the capacity of the decoder, we can systematically control the amount of detail that can be captured by the model. Due to our specific choice of dynamics family, an easy way to do so is by controlling the frequency of periodic activation functions. We use $\text{SIN}(\cdot)$ as activation due to the fact that $\omega \rightarrow 0 \Rightarrow \text{SIN}(\omega \cdot x) \rightarrow x$ (i.e. linear activation) and hence the model looses most of its capacity. Figure 7 shows that as we reduce activation function frequency, high frequency content is removed from the data in the reconstruction.

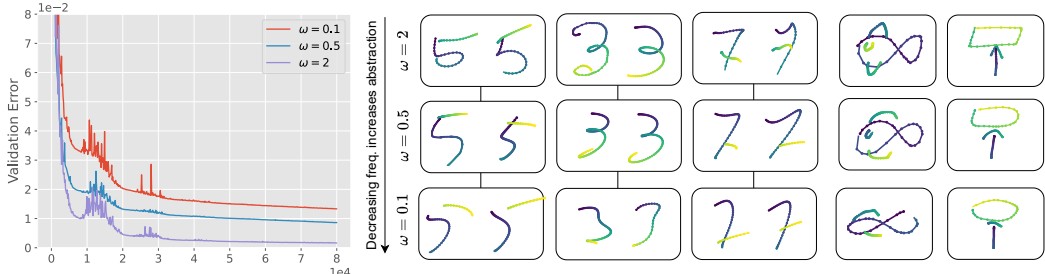

Figure 7: Illustration of SketchODE abstraction effect by varying activation frequency $\omega$. Decreasing the value of $\omega$, leads to increasing abstraction/reduced high-frequency detail.

## 6 CONCLUSIONS

In this paper, we proposed a fundamentally new paradigm for representing chirographic drawing data like handwriting, diagrams, and sketches. The SketchODE framework and its unique functional data representation posses several qualities that its discrete-time analogues do not: generation of inherently scalable arbitrary-resolution images, data efficient learning due to a built-in inductive bias towards continuity, ability to conditionally sample and perform 1-shot interpolation with a deterministically trained model. One drawback however is the computational complexity (approx 4-5× more computation time than discrete counterparts) of training which require computing higher order derivatives in both forward and backward pass. The constant-memory (w.r.t time horizon) nature of the training algorithm however, allows packing more samples in a batch for the same memory footprint. Despite computational challenges, this work may provide a conceptual primitive for modelling more complex but constrained structures (e.g. fonts, icon and vector arts). More generally, this is the first continuous-time Seq2Seq model, which may find application in diverse domains beyond chirography, like finance, audio or video, which are inherently continuous in nature.

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

## A  GENERATIVE LOSS FUNCTION

In order to provide extra flexibility to model variations, we can define an explicit density over decoded trajectories. We enforce distributions $q(\mathbf{z}|\mathbf{s}^{(i)})$ and $q(\Theta)$ over the latent code and the parameters of the dynamics respectively. With a conditional density on $\mathbf{z}$, we can treat the ODE trajectory as Continuous Normalizing Flow or CNF (Chen et al., 2018) that transforms the induced density $q(\mathrm{H}_0|\mathbf{z})$ through a continuous mapping. We sample from the conditional and prior by means of reparameterization (Kingma & Welling, 2014) and compute the full loss which includes another component $\mathcal{L}_{\mathrm{prob}}$ along with $\mathcal{L}_{\mathrm{fit}}$ accounting for the probabilistic form.

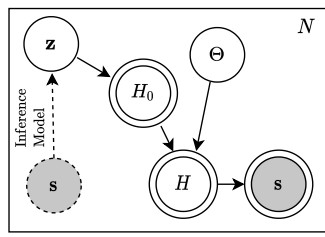

Figure 8: The generative model as probabilistic graph.

We first define the generative model $q(\mathbf{s})$ comprised of a global latent vector $\mathbf{z}$ with prior $p(\mathbf{z})$, the stochastic ODE parameters $\Theta$ with prior $p(\Theta)$ on all its parameters (i.e. ODE dynamics being a Bayesian Network) and the initial hidden state $\mathrm{H}_0$. The variational distribution is assumed to be factorized as

$$q(\mathrm{H}, \mathrm{H}_0, \mathbf{z}, \Theta|\mathbf{s}) = q(\Theta) \cdot q(\mathbf{z}|\mathbf{s}) \cdot q(\mathrm{H}_0|\mathbf{z}) \cdot q(\mathrm{H}|\mathrm{H}_0, \Theta)$$

Please refer to Fig. 8 for visual illustration of the dependency structure. The notations $\mathbf{s}$ and $\mathrm{H}$ (without time suffix) denote random variables that describe continuous time sequences within the time range $(t_0, t_1]$. Therefore, the *Evidence Lower Bound* objective denoted as $\mathcal{L}_{\mathrm{prob}}$ is given by

$$\mathcal{L}_{\mathrm{prob}}(\mathbf{z}, \Theta; \mathbf{s}^{(i)}) = -\mathrm{KL}\left[q(\mathbf{z}, \mathrm{H}, \Theta|\mathbf{s}^{(i)})\|p(\mathbf{z}, \mathrm{H}, \Theta)\right] + \mathbb{E}_{q(\mathbf{z}, \mathrm{H}, \Theta|\mathbf{s}^{(i)})}\left[\log q(\mathbf{s}^{(i)}|\mathrm{H}, \Theta)\right] \quad (6)$$

For deterministic mappings, we assume uninformative priors, i.e. the densities are one. So, $p(\mathrm{H}_0|\mathbf{z}) = p(\mathrm{H}|\mathrm{H}_0, \Theta) = 1$. Also, we are not defining any explicit observation model, i.e. $q(\mathbf{s}|\mathrm{H}, \Theta) = 1$. The following simplification can be done to Eq. 6 based on assumptions

$$
\begin{aligned}
\mathcal{L}_{prob} &= -\mathrm{KL}\left[q(\mathbf{z}, \mathrm{H}, \Theta|\mathbf{s})\|p(\mathbf{z}, \mathrm{H}, \Theta)\right] \\
&= -\mathbb{E}_{q(\mathbf{z}, \mathrm{H}, \Theta|\mathbf{s})}\left[\log \frac{q(\mathbf{z}|\mathbf{s})}{p(\mathbf{z})} \cdot \frac{q(\Theta)}{p(\Theta)} \cdot \frac{q(\mathrm{H}|\mathbf{z}, \Theta)}{p(\mathrm{H}|\mathbf{z}, \Theta)}\right] \\
&= -\underbrace{\mathbb{E}_{q(\mathbf{z}|\mathbf{s})}\left[\log \frac{q(\mathbf{z}|\mathbf{s})}{p(\mathbf{z})}\right]}_{\text{KL Divergence}} - \underbrace{\mathbb{E}_{q(\Theta)}\left[\log \frac{q(\Theta)}{p(\Theta)}\right]}_{\text{KL Divergence}} - \mathbb{E}_{q(\mathbf{z}, \mathrm{H}, \Theta|\mathbf{s})}\left[\log q_{\mathrm{CNF}}(\mathrm{H}|\mathrm{H}_0, \Theta) q_{\mathrm{NF}}(\mathrm{H}_0|\mathbf{z})\right] \\
&= -\mathrm{KL}\left[q(\mathbf{z}|\mathbf{s})\|p(\mathbf{z})\right] - \mathrm{KL}\left[q(\Theta)\|p(\Theta)\right] - \mathbb{E}_{\substack{\mathbf{z} \sim q(\mathbf{z}|\mathbf{s}), \Theta \sim q(\Theta) \\ \mathrm{H}_0 = \mathbf{L}_{dec}(\mathbf{z}) \\ \mathrm{H} = \mathrm{ODE}(\mathrm{H}_0, \Theta)}}\left[\log q_{\mathrm{CNF}}(\mathrm{H}|\mathrm{H}_0, \Theta) q_{\mathrm{NF}}(\mathrm{H}_0|\mathbf{z})\right]
\end{aligned}
$$

Restricting $\mathbf{L}_{dec}$ to be invertible, the log-density of the induced $q(\mathrm{H}_0|\mathbf{z})$ can be computed using change of variable formula as shown in Rezende & Mohamed (2015)

$$\log q_{\mathrm{NF}}(\mathrm{H}_0|\mathbf{z}) = \log q(\mathbf{z}|\mathbf{s}) - \log\left|\det \frac{\partial \mathbf{L}_{dec}}{\partial \mathbf{z}}\right|$$

The log density of the hidden trajectory can be computed using the instantaneous density formula for Continuous Normalizing Flow (CNF) derived by Yildiz et al. (2019)

$$\log q_{\mathrm{CNF}}(\mathrm{H}|\mathrm{H}_0, \Theta) \approx \sum_t \left[\log q_{\mathrm{NF}}(\mathrm{H}_0|\mathbf{z}) - \int_{t_0}^t \mathrm{tr}\left(\frac{d\mathcal{F}_\Theta}{d\dot{\mathbf{a}}}\right)\right]$$

Both $q(\mathbf{z}|\cdot)$ and $p(\Theta)$ are modelled with isotropic Gaussians. With a reparameterized sample from both, Eq. 6 can be computed and added to the regression loss. Note that all hidden states in above equation can be replaced with its augmented version since visible states are deterministic given other unobserved quantities.

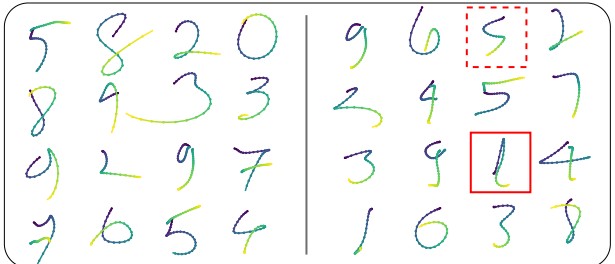

Figure 9: Original data from VectorMNIST on the left and unconditional samples on the right. Just like any VAE-based generative model, reconstruction is traded-off by posterior-matching objective. Sometimes, the model generates samples that are blend of two categories (dotted boxed; blend of $4$ and $5$), or samples that are entirely unrecognizable (solid boxed).

We experimented with a basic version of the generative model on VectorMNIST dataset and samples unconditional samples from the model. Please refer to figure 9 for visual examples.

## B  DATA-CONTROLLED DYNAMICS

A dynamical system can be controlled by a parameter that do not "evolve" over time but can be differentiated with. Such design is loosely inspired by SketchRNN (Ha & Eck, 2018) which concatenates decoder inputs with the global latent vector in order to prevent the recurrence from *forgetting* the initial state. With the dynamics function $\mathcal{F}_\Theta(\dot{\mathbf{a}}, t, \mathbf{z})$, we can easily achieve this in practice by solving the following dynamical system

$$\ddot{\mathbf{a}}(t) = \widehat{\mathcal{F}}_\Theta\left(\begin{bmatrix}[\dot{\mathbf{a}} \ t]^T \\ \mathbf{z}\end{bmatrix}\right) = \begin{bmatrix}\mathcal{F}_\Theta(\dot{\mathbf{a}}, t, \mathbf{z}) \\ \mathbf{0}\end{bmatrix}$$

The fixed zero vector as second derivative of $\mathbf{z}$ prevents it from co-evolving along with $\dot{\mathbf{a}}$. Since $\mathbf{z}$ is now a part of the state, Adjoint Backpropagation (Chen et al., 2018) naturally allows easy computation of the partial derivative.

## C  MULTI-STROKE FORMAT

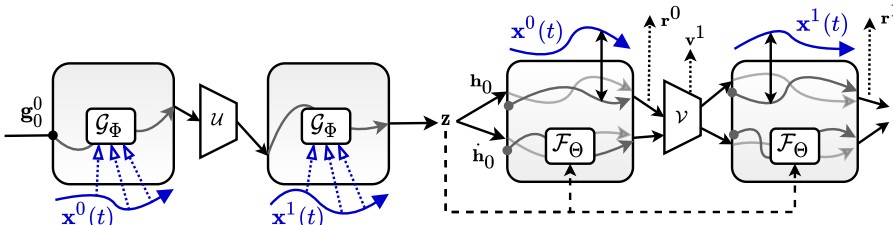

Figure 10: SketchODE variant for multi-stroke format.

Unlike full-sequence format, we use the architecture defined in the main paper for processing one single stroke at a time while sharing the parameters across different strokes. Refer to Fig. 10 for a concise diagram.

For CDE based encoder, a sequence of strokes $\{\mathbf{x}^1(t), \mathbf{x}^2(t), \cdots\}$ is processed individually with a CDE with initial state $\mathbf{g}_0^k := \mathbf{g}^k(t = t_0)$ and final state $\mathbf{g}_1^k := \mathbf{g}^k(t = t_0)$. The time range for each stroke, if not available in dataset, can be fixed as a hyperparameter. Initial state for the $k^{th}$ stroke can be computed by projecting the last state of $(k-1)^{th}$ stroke with parametric transform $\mathcal{U}$ (e.g. MLP)

$$\mathbf{g}_0^k = \mathcal{U}(\mathbf{g}_1^{k-1}), \text{ where } \mathbf{g}_1^{k-1} = \mathbf{g}_0^{k-1} + \int_{t_0}^{t_1} \mathcal{G}_\Phi(t, \mathbf{g}) \, \dot{\mathbf{x}}^{k-1}(t) \cdot dt$$

The initial state of the very first stroke is always fixed to a constant, e.g. $\mathbf{g}_0^0 = \mathbf{0}$.

For ODE based decoder, a similar strategy is followed with the exception that the initial state of the first stroke is set to $A_0$ where $\dot{\mathbf{s}}_0 = \mathbf{s}_0 = 0$ and $H_0 = \mathbf{L}_{enc}(\mathbf{z})$. The initial state of the $k^t h$ stroke is produced by transforming the last state of $(k-1)^{th}$ stroke with another parametric transform $\mathcal{V}$ (e.g. MLP)

$$H_0^k = \mathcal{V}(A_1^{k-1}), \text{ where}$$

$$A_1^{k-1} = A_0^{k-1} + \int_{t_0}^{t_1} \begin{bmatrix} \dot{\mathbf{a}}(t) \\ \mathcal{F}_\Theta \end{bmatrix} dt$$

Please note that for every stroke, the initial position and velocity is fixed to zero. Such a strategy leads to the ODE dynamics learning *shared* stroke model. It follows earlier works like Das et al. (2020; 2021); Aksan et al. (2020) and require a starting position $\mathbf{v}^k$ for every stroke to be predicted separately, which can be regressed during training. Also, an *end-of-sample* flag is necessary for terminating the inference which can be achieved by defining a random variable $\mathbf{r}^k \in \{0, 1\}$ as whether the $k^{th}$ stroke is the last stroke and predicting the probability of it being one

$$p(\mathbf{r}^k = 1 | A_1^k; \theta) = \text{Sigmoid}(\mathcal{R}_\theta(A_1^k))$$

## D  PRACTICAL DATA STRUCTURES & MINI-BATCHING

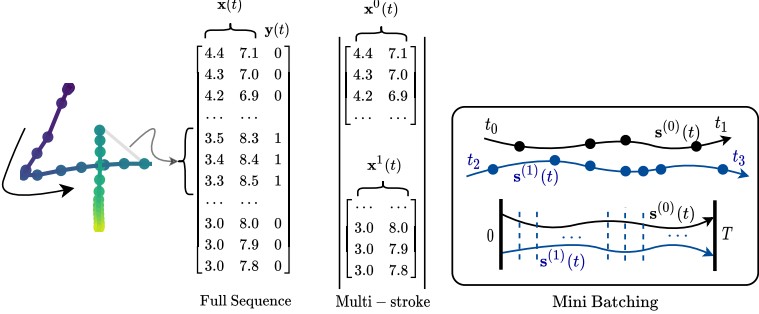

Figure 11: Visual illustration of data formats, resampling and mini-batching strategy.

Samples in full-sequence data format is represented by a length $N$ sequence $[(t^j, \mathbf{s}(t = t^j))]_{j=1}^N$ where $t^j \in [0, T]$ are non-uniform time-points. For multi-stroke format, each data sample may consists of multiple strokes each represented as a discrete coordinate sequence. The end coordinate of the previous stroke and starting coordinate of the following stroke are connected via straight line and $\mathbf{y}(t = t^j)$ is set to 1 for all time points on it; otherwise $\mathbf{y}(t = t^j) = 0$. Please refer to Fig. 11 for visual illustration. For more time-sensitive applications, the time-range can be left as it is in the raw data and can be regressed from the latent vector (full-sequence format) or initial decoder state (multi-stroke format). At inference, given a latent code, the decoder ODE can be solved using the predicted time-range.

In practice, every sample in the dataset needs to have a format shown in Fig. 11. For full sequence format, every stroke including the pen-up strokes (i.e. joining line between previous stroke-ending and next stroke-begining) are kept in order. The pen-up stroke is shown in gray and the corresponding $\mathbf{y}(t)$ is set to one. The multi-stroke format do not require any $\mathbf{y}(t)$. All samples are spatially scaled down inside the unit circle, i.e. $\|\mathbf{x}(t)\|_2 \leq 1 \, \forall t \in [0, T]$. For datasets with high temporal sampling rate (e.g. DiDi and *Quick, Draw!*), we reduce it by Ramer–Douglas–Peucker algorithm (Douglas & Peucker, 1973) as a preprocessing step.

Efficient training of Neural ODE/CDE models further require mini-batching, which is different than traditional models. For samples with different number of time-points, we create a common sequence of time-points by *union*-ing individual time-point sequences. An ODE solution trajectory is then evaluated at the time points in the union and masking is done to discard irrelevant time points for each sample. For even more efficiency, we can re-sample each data in a mini-batch at a fixed sequence of uniform time-points. This is achieved by linear interpolation of points along time. Please refer to Fig. 11 for visual illustration.

## E  VISUAL SAMPLES

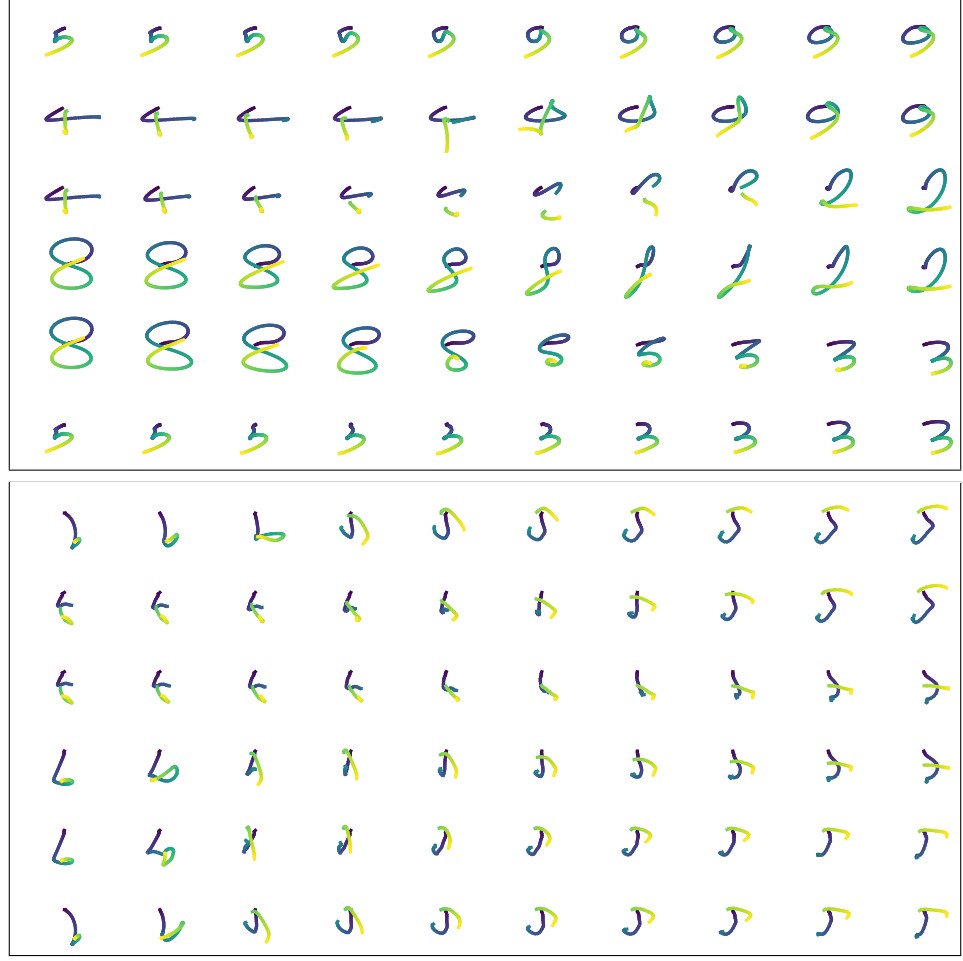

Figure 12: Interpolation on latent space by cyclic walk through 6 samples of different categories. SketchODE (above) is far better with smooth animatic interpolation while RNN-RNN not only fails to faithfully reconstruct but also suffers from severe mode collapse.

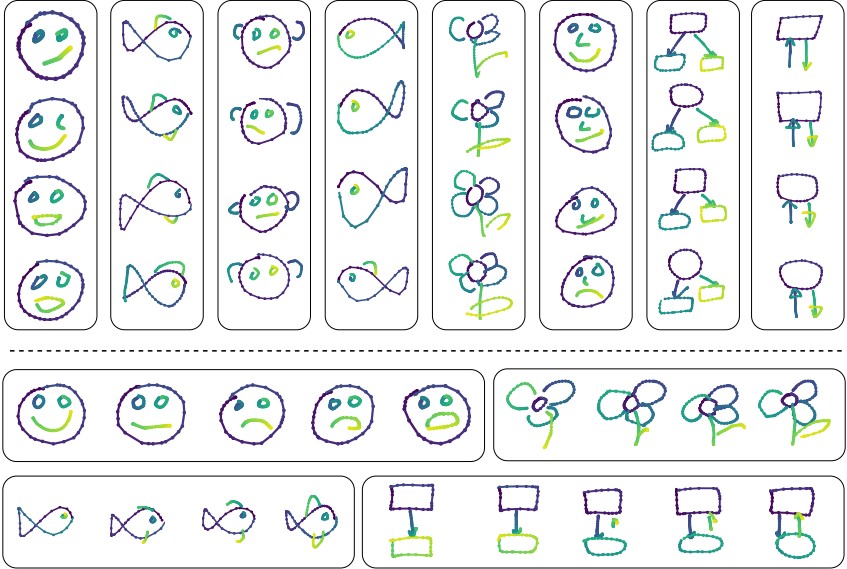

Figure 13: Some examples of conditional sampling (above) and animatic interpolation (below). For conditional sampling, the samples are conditioned on the sketch in the first row. Each of the interpolation examples is to be viewed left to right.

