# OpenReview forum: "SketchODE: Learning neural sketch representation in continuous time"
_ICLR.cc/2022/Conference — ICLR 2022 Poster_

### Official Review · Reviewer_KvGm · 2021-10-26

**Correctness:** 3
**Technical Novelty And Significance:** 2
**Empirical Novelty And Significance:** 2
**Recommendation:** 5
**Confidence:** 4

**Main Review:**

The authors propose an approach of representing online handwriting data in the continuous space. One of the central claims made by the authors is that models that use discrete representation need to spend their learning capacity for learning both the global structure and the temporal continuity (basically, the sampling method), while the models using continuous representation can use their full capacity to represent the global structure only, thus becoming more data-efficient.

Authors represent the continuous state of the pen $s(t)$ (which can include either only coordinates, if a single stroke is encoded, or both the coordinates and the pen state, if several strokes are encoded together) as the observations following the hidden process h(t). Authors also capture the "velocity" of the pen tip in their model. To obtain the ground truth data for it, they fit spline curves to the existing data points.

The encoder of the proposed model used Neural CDEs and computes the hidden state by integrating the input trajectory with a kernel that is a parametrized neural network. The decoder of the proposed model uses Neural ODEs to model the velocity, which can then be turned into the recostruction of the trajectory itself. Both during encoding and decoding, the forawrd pass involves computing an ODE solution, which is done by the black-box ODE solver, using Range-Kutta method. The loss is a point-wise loss (Huber loss or smooth L1) between the points of real and recostructed trajectory, with different sampling frequencies according to the dataset. Authors provide extensive details on the training procedure.

Authors evaluate their approach on QuickDraw dataset of handwritten sketches, DiDi dataset of handwritten diagrams, and the collected small-scale dataset of vectorized MNIST digits. Authors suggest a data augmentation technique to increase the amount of training data, based on adding continuous noise to the continuous trajectory, rather than by adding independent Gaussian noise to the points.

Authors perform the following evaluations:
* Accuracy of the trained recognition CNN on the reconstructed data from 4 models: RNN-RNN seq2seq model trained on raw points data, the one proposed by the authors (SketchODE), and two others: CoSE which encodes a discrete sequence into an implicit representation from which the points of the recostructed curve can be sampled by providing a $t\in[0,1]$, and BezierSketch, which operates on Bezier curves parameters as inputs / outputs to the autoencoder model. CoSE achieves the highest accuracy, closely followed by RNN-RNN, and SketchODE.

**Concern** The Table 1 describing the recognition accuracy of the classifier trained on the real samples, on the reconstructed samples, is meaningless without knowing what is the recognition accuracy of the said classifier on the real data. If it is, for example, lower than 93%, that means that the autoencoder models suffer from a mode collapse and are only able to generate some modes of the training data distributions, that are more recognizable, than the whole distribution in general.

* Authors perform visual comparison to RNN-RNN in terms of the quality of reconstructions, data efficiency, and the ability to interpolate in the latent space. They show favourable comparisons in all cases.

**Concern** Authors compare to the model that operates on the discrete sequence of points, but ignore the comparison to BezierSketch, and, more importantly, CoSE, which are able to provide the continuous reconstructions, and which would have been much fairer comparisons.

**Concern** Authors talk about data efficiency, but don't mention whether they use the same data augmentation when training RNN-RNN (adding Perlin noise), or is that only applicable when training SketchODE (and, if that is the case, whether an independent Gaussian noise data augmentation is applied to RNN-RNN). Authors also don't mention the effect of the data augmentation on the performance of their model.

**Concern** Authors show  the interpolation in latent space between two novel objects, and compare the quality to RNN-RNN. While this indeed indicates the smoothness of the learned latent space, but, (1) this is incomplete without comparisons to methods that allow outputting a continuous curves, and (2) interpolation in the vector space of curves would have likely produced the results similar to the ones obtained by the authors.

Authors show an interesting emergent property of their work, namely, by controlling the capacity of the model decoder, they can control the amount of "complexity" or "high frequency" in their reconstructions.


**Summary Of The Paper:**

The paper describes a way of representing online handwriting data, typically seen as a discrete sequence, in a continuous space, using neural ODEs. The learned representation allows sampling from and interpolation in the latent space. The results in the paper are compared (mostly qualitatively) to autoencoder-learned representations based on representing data as discrete sequences, Bezier curves, and implicit surface representation from differentable geometrty.

**Summary Of The Review:**

Authors propose a way of encoding online handwriting data in a continuous space representation by using Neural ODEs, and show interesting properties of the learned model. This is a novel idea interesting to wider audience.

Their main weakness of the paper, however, is the experimental section, and based on its' current content I can not recommend the paper for acceptance.
* Most importantly, the comparisons are only done to an approach that operates purely on the discrete sequences of points, rather than the ones that, similar to the approach outlined by the authors, allow sampling a continuous trajectory as the output of their decoder (BezierSketch, CoSE), and whether the provided approach provides capabilities beyond those is an open question. Combined with the complexity of the training requiring ODE solvers, it is not clear whether the proposed approach provides any advantages.
* The ablation study is virtually non-existent and does not allow to understand the effects of some of the decisions made by the authors: how does continuous data augmentation affect the quality of the results, rather than the validation error? What are the trade-offs between using a single-stroke representation vs multiple strokes representation? How does the multiple strokes model behave with sequences containing longer than 5 strokes, and how does a single stroke model handle long strokes?

---

> ### Author Response · Authors · 2021-11-22
> **Official response to reviewer KvGm (PART 1)**
>
> Thank you for taking time and writing an elaborate review with a great summary of the paper and raising a few concerns. We clarify most of your concerns below with some agreements and disagreements.
>
> **Clarifications:** Before answering your specific concerns, we would like to clarify on few comments you made.
>
> 1. "*To obtain the ground truth data for it, they fit spline curves to the existing data points*". This isn't true per se. We don't use splines to obtain ground truth. The only place splines were used was in the CDE-based encoder, which uses splines to get *better derivatives*. The decoder however, is entirely free of any such spline fitting - `Eq. 5` operates entirely on discrete points. By the way, the encoder can also be implemented with numerical derivatives with only a slight loss of accuracy.
>
> 2. "*Single-stroke model*". Please note they aren’t really “single-stroke” - they are one functional representation for the entire sketch which may contain multiple strokes. We believe and assume that this is what you meant.
>
> **High-Level Points:** Apart from the above two minor clarifications above, we would also like to reiterate a few high-level points that we believe will clarify some doubts and create a context for the rest of our response.
>
> SketchODE is not primarily a generative model, rather a deterministic autoencoder (although we did provide a reasonable extension to a generative model in `Appendix A`). The inherent properties of our deterministic model  (sampling & interpolation) that are reminiscent of typical generative models are consequences of two properties of (neural) ODEs: 1. Deterministic & 2. End-to-End continuous nature. We therefore report comparisons with their deterministic counterparts (`Section 5.2` end of first paragraph says ".. trained deterministically"). The goal of the paper, as correctly reiterated and appreciated by other reviewers, is to show that truly continuous-time sequence models using ODEs can possess a number of interesting properties even in deterministic settings. We also emphasize that there are two distinct but related concepts of "continuity” in our setup: **[A]** Temporal continuity, i.e. the solution trajectory is continuous w.r.t time or equivalently, $\frac{ds(t)}{dt}$ is well behaved. **[B]** Continuous mapping from latent to solution trajectory, i.e. $\frac{d\mathbf{s}(t)}{d\mathbf{z}}$ (known as “sensitivity” in ODE literature) is well-behaved. The reason sampling/interpolation properties would emerge in a deterministic ODE-ODE continuous-time system (like ours) has more to do with the later and less with the former.
>
> **Comparison with continuous-stroke based models (concern #2 & #4):**
>
> The comment about ".. authors ignore comparisons with BezierSketch .. CoSE .." isn’t quite accurate. We did compare our method with deterministic versions of them in terms of reconstruction performance in `Table 1` and found that our method is competitive but not best - however, we did find some compelling properties we wouldn’t get in competing methods.
>
> We would like to clarify on your worry that "BezierSketch [1] / CoSE [2] .. have likely produced similar results ..". Please recognize the fact that BeizerSketch/CoSE may have continuous strokes, but their relational models (`Eq. 2` of [2] and `Eq. 11` of [1]) are still autoregressive (AR) in nature. Any deterministic model with a "latent to AR sequence" structure cannot have [B]-type continuity (reconfirmed by reviewer `#zrrF`), even if it has [A]-type continuity (piecewise; as in [1,2]). So, any perturbation in the latent vector would lead to a breakage of its relational structure and distort the geometry of strokes **even if they remain continuous in time**. The elegance of ODE based sequence models for chirographic data is that it can have **both [A] & [B]-type continuity** while modelling the entire sketch including pen-up events. It does so by packing both the "*stroke-model*" (parameters of the dynamics function) and "*relational-model*" (event functions $\mathcal{U}$ and $\mathcal{V}$) into one unified structure with end-to-end differentiability.
>
> We did a small experiment where we perturb the latent vector with similar gaussian noise for all three competing models and check the recognition accuracy (in %) of the generated samples with the same classifier CNN (mentioned in `Section 5.2`).
>
> |% Accuracy | BezierSketch | CoSE | SketchODE |
> |---|---|---|---|
> |VectorMNIST | 93.06 | 97.85 | 97.69 |
> |QuickDraw | 83.22 | 81.43 | 92.20 |
> |DiDi | 80.15 | 87.27 | 95.13 |
>
> We noticed that noise perturbed samples are not as recognizable for CoSE/BezierSketch as in SketchODE. Unsurprisingly, we found that this is due to the recurrence breaking up the relational structure, especially in datasets with higher number of strokes. Please refer to the updated `Fig. 5` in the paper.
>
> .. to be cont'd .. (Please move on to PART 2)

---

> ### Author Response · Authors · 2021-11-22
> **Official response to reviewer KvGm (PART 2)**
>
> .. cont'd .. (Please read PART 1 first)
>
> **Suspecting mode-collapse (concern #1):**
>
> There is no mode collapse. Please note that mode-collapse as such is a failure mode of generative models, and our SketchODE is a deterministic autoencoder for which this is not an applicable failure mode. The closest thing to mode collapse for a deterministic AE would be to always output a few memorized examples without faithfully autoencoding the input. If this happened the reconstruction error on the validation set would be high, but we can see this is not the case in `Table 1`. Therefore there is nothing like mode collapse.
>
> **Data augmentation in SketchODE vs Baseline (concern #3):**
>
> We use Perlin Noise only with our method since we proposed it.RNN-RNN baseline uses standard independent jittering and we didn’t want to alter the baseline. On your request, we tried a small experiment on VectorMNIST by training an RNN-RNN model with Perlin noise augmentation and found out that it improves the validation error only by a small margin (in the order of $10^{-4}$), in contrast to about $4*10^{-4}$ for SketchODE (refer to `Fig 5`).
> *More importantly*, we emphasize that the augmentation still does not affect the more fundamental difference in inherently continuous sampling between SketchODE and RNN-RNN. To illustrate this, we conducted the same experiment as in the table in the first part of this response (recognition accuracy of the generated samples) but for the RNN-RNN model trained with "Discrete Jittering" (called RNN-Baseline) and "Perlin noise" augmentation (called RNN-Perlin). We found that RNN-Baseline and RNN-Perlin had $74.34\%$ and $78.56\%$ accuracy of recognition which is still not comparable to SketchODE (97.69%). In summary, the Perlin noise may help discrete RNN models in reconstruction, but the continuity is still broken when latent space is sampled.
>
> **Other concerns:**
>
> 1. **How does augmentation affect the quality, rather than validation error?** We would like to reiterate the fact that our model is a reconstruction driven autoencoder and hence validation reconstruction performance is a sufficient indicator for its performance. Any reduction in "quality" would show up as reduced validation error.
>
> 2. **Trade-offs between full-sequence and multi-stroke:** Yes, full-sequence and multi-stroke models do have a trade-off between representational power and training dynamics. We initially noticed that training the "full-sequence" model with relatively long sequences is quite hard, and switched to the "multi-stroke model" for complex datasets (mentioned in implementation details). Multi-stroke model is quite flexible and resembles CoSE, as it decouples the underlying shared stroke-model (ODE dynamics network) and relational model (the state-update/event functions $\mathcal{U}, \mathcal{V}$). For a quick quantitative analysis (on VectorMNIST), we would like to share that under the same experimental settings described in `Section 5.1`, a full-sequence model takes $\sim 20k$ iterations to converge to converge (to a validation error reported in `Table. 1` of the paper) whereas a multi-stroke model takes $\sim 10-12k$ iterations. However, the multi-stroke model consumes more walltime per iteration ($\sim 2x$ than full-sequence with batch size 256). So the total walltime remains fairly the same. For complex datasets, however, multi-stroke is the only option.
>
> 3. **Computational complexity:** Please follow the discussion with reviewer `#zrrF`.
>
> ---
>
> [1] Ayan Das, Yongxin Yang, Timothy Hospedales, Tao Xiang and Yi-Zhe Song. "BezierSketch:  A generative model for scalable vector sketches". ECCV, 2020.
>
> [2] Emre Aksan, Thomas Deselaers, Andrea Tagliasacchi and Otmar Hilliges. "CoSE: Compositional stroke embeddings". NeurIPS, 2020.

---

> > ### Comment · Reviewer_KvGm · 2021-11-24
> > **Reply to the authors response**
> >
> > I thank the authors for the detailed replies. They have cleared the part of the concerns I have, namely the data augmentation effect, and the trade-off between full-sequence and multi-stroke. I also accept the argument that for the full-sequence models varying the latent space is qualitatively different for SketchODE and other approaches, since SketchODE  models the whole sketch continuously. In particular, the newly added Figure 13 shows the power of the model in this scenario, and I appreciate the authors adding it. In view of the authors responses, I upgrade my rating to 5.
> >
> > Authors have themselves said that full-sequence format is only applicable to small datasets - which answers my question about the performance of the full-sequence model on inputs larger than several strokes.
> >
> > For a multi-stroke format, where each individual stroke is encoded separately, I still imagine that interpolation in the latent space would not be visually different for the proposed model and the models having continuous outputs.

---

> > > ### Author Response · Authors · 2021-11-27
> > > **Thank you. Responding to further doubt about the multi-stroke model.**
> > >
> > > Thank you very much for appreciating our response, and for increasing the score.
> > >
> > > We would like to shed light on the one remaining doubt mentioned in your latest comment, i.e. that the multi-stroke SketchODE will behave similarly to [1,2] in terms of latent space interpolation. Actually, the multi-stroke SketchODE behaves like the full-sequence SketchODE and not like [1,2] in terms of smooth latent space interpolation. (See `Fig. 5` and `Fig. 6`, where our latent space interpolation is smooth despite use of multi-stroke format in DiDi and QuickDraw).
> > >
> > > Let us explain why. Recall the notations for denoting each stroke $\mathbf{x}^k$ and the continuous hidden state $\mathbf{h}^k(t)$ as mentioned in `Appendix C`.
> > >
> > > **What happens in the relational model of [1] and [2] at inference ?**
> > >
> > > Relational models for sketch uses specific embedding $\mathbf{\lambda}^k$ for each stroke $\mathbf{x}^k$ : for [1] it's native Bezier representation and for [2] it's a custom embedding learned by means of differential geometry. A recurrent model is then used to predict
> > >
> > > $\mathbf{\lambda}^{k+1}, \mathbf{h}^{k+1} = \mathrm{RNN}(\mathbf{\lambda}^k, \mathbf{h}^k)$
> > >
> > > Deterministic RNNs may fail to keep the [B]-type continuity due to their "output from the previous step being fed into the next step as input". If any step $k$ fails to predict its true target $\mathbf{\lambda}^k_{\mathrm{true}}$ by an error $\mathbf{e}^k$, the recurrence *amplifies* the error in later time-steps as
> > >
> > > $\mathbf{\lambda}^{k+1}\_{\mathrm{true}} + \mathbf{e}^{k+1}, \mathbf{h}^{k+1} = \mathrm{RNN}(\mathbf{\lambda}\_{\mathrm{true}}^k + \mathbf{e}^k, \mathbf{h}^k)$
> > >
> > > Depending on the nature of the RNN’s state-to-output mapping, this error can amplify significantly as it unrolls in time.
> > >
> > > **What happens in a multi-stroke ODE decoder ?**
> > >
> > > Our multi-stroke ODE decoder model does not have any external inputs throughout its entire evolution of state. Once the latent vector is decided, the whole sequence (including pen up events) is fixed. Rather than predicting local structure one step at a time, our ODE based decoder predicts the full (global and local) structure all at once.
> > >
> > > The reason the "*latent to sketch"* mapping for the multi-stroke decoder still has inherent continuity despite having separate strokes is due to its reliance on MLPs $\ \mathcal{U}$ and $\mathcal{V}$. Since the final state of previous stroke $\mathbf{g}^{k-1}_1$ is mapped to initial state of next stroke $\mathbf{g}^k_0$ through a continuous map, any infinitesimal change in the (last state of) previous stroke gives rise to an infinitesimal change in the (first state of) next stroke. This way, the entire continuity is preserved starting from latent vector to the end stroke.

---

### Official Review · Reviewer_zrrF · 2021-10-28

**Correctness:** 4
**Technical Novelty And Significance:** 4
**Empirical Novelty And Significance:** 4
**Recommendation:** 8
**Confidence:** 3

**Main Review:**

Overall, I think this is a strong paper which introduces an interesting model and demonstrates various compelling properties of this model through a series of experiments on several handwritten datasets. The main strengths and weaknesses (in my eyes) are described below.

**Strengths**:
- I think the main strength of the paper is the demonstration of the various compelling properties that arise from modeling the handwritten data continuously. In particular, I really like the experiments showing that by simply adding noise to the latent vector we can generate conditional samples from the model, due to the continuous nature of the setup (which would break down in the autoregressive RNN case). The latent interpolation experiments are also very nice. The one shot learning capabilities demonstrated on VectorMNIST are also compelling.
- The model and ideas are very well motivated. Handwriting and drawing are inherently continuous, so it makes sense to model this data continuously. The authors do a good job of demonstrating this, both by contrasting it with previous work and by showing through experiment that this works well.
- The paper is generally well written and the figures are great.
- While the model is somewhat complicated and has lots of moving parts, each part is generally quite well motivated.
- The experiments are generally thorough and well done (on 3 fairly different datasets). The visualizations of the results are great. However, as discussed in the weaknesses section, it would be nice to have more samples/examples from the model (particularly on the more complicated datasets), as well as having error bars.

**Weaknesses**:
- It would be nice to have more samples and results from the model in the appendix. The majority of the results both in the main paper and appendix are shown on VectorMNIST. It would be nice to have e.g. more examples of latent interpolations and conditional sampling on Quick Draw and DiDi. At this point, since there are so few examples they can feel cherrypicked.
- The authors do not discuss the large amount of relevant work that has been done around using neural ODEs for physics (see Hamiltonian neural networks and Lagrangian neural networks and the numerous follow up works since then). Sentences like “there has been little applied work using neural ODEs” and “parameterized ODE dynamics models have largely been treated as a replacement for ResNets where the intermediate states are of little importance” are misleading. As far as I can tell, the authors do not cite a single neural ODE + physics paper which both use the intermediate states and are applications of neural ODEs.
- The authors overclaim their contributions in certain parts of the paper. This is already a great paper, so it is not necessary to do this. For example, “we increase the flexibility of Neural ODE models by introducing a new class of parameterized dynamics functions”. As far as I can tell the authors simply replace tanh/relu MLPs with SIREN (a pre-existing model), which effectively corresponds to changing an activation function. The properties of SIREN models are already well described in other papers, so I think it is a stretch to call this “introducing a new class of parameterized dynamics functions”. In addition sentences like “Learning meaningful representations for chirographic drawing data such as sketches, handwriting, and flowcharts is a gateway for understanding and emulating human creative expression” seem unnecessarily grandiose and don’t provide much information.
- It would be nice to have a more thorough discussion of limitations. As far as I can tell, the only discussion of limitations is one sentence in the conclusion mentioning the computational complexity of the method. It would be nice to have some more thorough discussion of this or ideally a quantitative analysis of how much slower/expensive it is.
- On page 1, the notation h[n] and s[n] is used without being introduced. It only becomes clear what this is later in the paper.
- There are quite a few typos in the paper. I think it would be worth going over the paper another time to clean these up.

**(Some) Typos**:
- Page 3, top: “as generic functions have not yet been” -> “as generic functions have not yet been”
- Page 3, top: “Neural ODE framework” -> “The neural ODE framework”
- Page 3, bottom: “with Neural ODE as in Eq 1” -> “With a Neural ODE as in Eq 1”
- Page 4, middle: “computing the forward pass require” ->  “computing the forward pass requires”
- Page 4, middle: “a non-causal model as encoder” -> “a non-causal model as an encoder”

*Note*: While I am quite familiar with neural ODEs, I am less familiar with the literature around learning representations for handwritten data, and have therefore put a confidence score of 3 on the review. The authors seem to provide a thorough discussion of related work for handwritten data, but it is possible that there are papers that have been missed in that area which I do not know about.


**Summary Of The Paper:**

This paper proposes a new model, called SketchODE, for learning representations of sketches using neural ODEs. Specifically, the authors parameterize hand drawn strokes as solutions of ODEs and build an auto encoder like framework for learning the vector fields of these ODEs from data.

The decoder is modelled with a second order neural ODE acting on an augmented state (which allows for self overlapping trajectories which is necessary for handwritten data) that includes the stroke itself as well as an underlying hidden state. The decoder is made conditional by passing a latent vector to the model, which is mapped both to the initial state of the ODE as well as the dynamics function of the ODE. Varying this latent vector and solving the ODE then gives rise to different handwritten patterns.

The encoder is parameterized by a neural CDE which takes as input some ground truth trajectory and returns a new state which is mapped to the latent vector. The model is then trained end to end using a reconstruction loss between the ground truth and reconstructed trajectories.

In addition, the authors show that two additional tricks can further improve performance: using sine activation functions in the MLPs parameterizing the vector fields as well as using perlin noise to create (continuous) augmentations of the data.

The authors then discuss and demonstrate various compelling properties of their model. This includes the existence of a latent space (which allows for interpolation of handwritten data) as well as one shot learning capabilities for new classes of data.

The main contributions of the paper in my eyes are then:
- Introducing an interesting continuous neural model of handwritten data
- Demonstrating compelling properties of this continuous model over traditional discrete models
- Introducing the VectorMNIST dataset, which I believe could be useful for other researchers
- Interesting experiments on various handwritten datasets




**Summary Of The Review:**

This paper introduces a new continuous time model for learning representations of handwritten data. The paper is well motivated and well written and demonstrates various compelling properties of the proposed model through thorough experiments. While the paper has a few minor weaknesses, I believe it deserves to be accepted.

---

> ### Author Response · Authors · 2021-11-22
> **Official response to reviewer zrrF**
>
> Thank you very much for appreciating the novelty and taking your time to thoroughly evaluate the paper. We have fixed the typos in the updated version.
>
> **Citing ODE papers about Physics:** Thank you for pointing out this specific body of work. Although we were aware of it, we did not consider mentioning them due to the very different nature of domains. But we agree that it is a fair point. We have added some key papers in the related work section and toned down the statements we made about ".. little applied work .. ".
>
> **Overstatement about "new class of dynamics functions":** You are right that the only change that was made was replacing $\mathrm{ReLU}/\mathrm{Tanh}$ with $\mathrm{sin}/\mathrm{cos}$ in the activations of dynamics function. However, this change indeed introduces a different "class of dynamics functions" as we can now represent a very different set of vector-fields (as seen in `Fig. 4`) which is very hard to learn without it. So, we do not really think this is an overstatement. We did cite SIREN at the right place and claimed that we are "inspired" from it. But our claim is focused on our contribution of introducing *temporal frequency* content in dynamics functions, while $\mathrm{sin}/\mathrm{cos}$ activations were used by SIREN to model pixel level a.k.a *spatial frequencies*. We included a few additional words in `Section 4.5` for improving the clarity on the scope of our claim.
>
> **Computational complexity (Limitation):** Yes, all the emergent properties of our framework do incur a relatively higher computation cost. However, it isn't so bad. As Neural ODEs are $\mathcal{O}(1)$ memory in terms of sequence length (i.e. no. of state evaluations), we can pack more samples in a batch while still under the memory budget. We also noticed (known earlier too) ODE models can tolerate high learning rate (notice our learning rate is quite high) and converge only in a slightly higher number of epochs as RNNs. Each epoch however takes more wall-time. Given enough parallelization resources, that can be mitigated too. In practice, for QuickDraw dataset with the same batch size (128), it takes $\sim 30$ mins and $\sim 2.5$ hours for RNN and SketchODE respectively to achieve the same rate of error. With increased batch size for SketchODE (2x batch size, more throughput, similar memory requirement) we can reduce it to $\sim 1.5-2$ hours. For comparison between two data-formats, please follow our response to reviewer `#KvGm`. We will include these in the final version.
>
>
> **More examples:** Thanks. We have included more in the updated version.

---

> > ### Comment · Reviewer_zrrF · 2021-11-25
> > **Thank you for your response!**
> >
> > Thank you for your response!
> >
> > I appreciate the updates you have added to the paper, including the extra samples in the appendix as well as the updated discussion around applications of neural ODEs. I also appreciate the results you share around computational complexity and runtimes - adding this to the paper will situate the model much better.
> >
> > Thank you also for adding a few additional words to Section 4.5. However, I still believe the statements are misleading. E.g. "Inspired by SIREN (Sitzmann et al., 2020), which introduced a way to increase spatial frequency content, we propose to use periodic activation functions in order to increase the temporal frequency content of solution trajectories". The original SIREN paper already has experiments on increasing temporal frequency, for example in the experiments on audio (where time is the only variable) and video (where time is one of the variables). Further, a large number of follow up works have also applied SIREN to both audio and video, so using SIREN for increasing temporal frequency is already well established. However, I have not seen SIREN being used with neural ODEs before, so it is fair to claim this as being novel.
> >
> > Thanks again for your response!

---

> > > ### Author Response · Authors · 2021-11-27
> > > **Thank you.**
> > >
> > > Thanks for the pointers on the usage of SIREN’s periodic activations. We will further clarify the claim accordingly in the final version.

---

### Official Review · Reviewer_S7jh · 2021-11-02

**Correctness:** 4
**Technical Novelty And Significance:** 3
**Empirical Novelty And Significance:** 4
**Recommendation:** 8
**Confidence:** 3

**Main Review:**

### Strengths
- The newly proposed continuous-time generative model is novel, and several intriguing observations are made. Although the chirographic data itself might have limited application, it served as a great benchmark for testing such a continuous-time generative model.
- The paper is clearly written and well organized overall.
- Considerations of the application on different data domain makes the implication broader.

### Weaknesses
- The tradeoff between computational complexity and representation power induced by using Neural ODEs is not directly handled. To mitigate this, the authors might want to simply share some information about the train/prediction cost and compare them with existing methods.
- For the sinusoidal activation functions, the frequency needs to be manually selected.

### Questions
- In the context of the multi-stroke dataset in Appendix C, what exactly are linear transformations u, v? Are they learned?

### Minor typos found
- At the end of the Introduction part: A surprising 'properly' of Seq2seq... -> property
- Section 4.2: 'Nautral' Spline curve -> Natural
- Section 5.1: 'Range'-Kutta method -> Runge


**Summary Of The Paper:**

By introducing the techniques from neural ODEs on chirographic data, the authors were able to train a first continuous-time generative model. New observations that are only available on continuous-time generative models are made.

**Summary Of The Review:**

This work successfully proposed the first continuous-time generative model and discovered intriguing observations. The possibility of different applications might interest a wide range of ICLR audiences, and hence I recommend accepting the paper.

---

> ### Author Response · Authors · 2021-11-22
> **Official response to reviewer S7jh**
>
> We thank you for recognizing the strengths of our method. We have fixed all typos in the updated version.
>
> **Frequencies need to be manually selected:** It is not necessary to select frequencies manually (we mentioned that they ".. can be learned .." in `Section 4.5`). Since sinusoids like $\mathrm{sin}(\omega \cdot x)$ and $\mathrm{cos}(\omega \cdot x)$ are both continuous and differentiable w.r.t $\omega$, we can treat them as learnable parameters. In fact, we did experiment with learnable frequencies (even individual $\omega$ for every layer) and found out they work quite well and mostly converge around $2 < \omega < 2.5$. Since we did not find any specific advantage over fixed $\omega$, we decided to treat them as hyperparameters.
>
> **What are $\mathcal{U}$ and $\mathcal{V}$ ? Are they learned ?** Yes. Sorry we missed to mention that they are simple parametric transforms like MLPs. Although `Appendix C` terms it as "linear transform", they can be non-linear for complex datasets with difficult-to-model relational structure. We have clarified this in the updated version. Theoretically, these transforms help the evolving state of an ODE $\mathbf{h}(t)$ change instantaneously - they are known as "events" [1] in ODE literature.
>
> **Computational complexity:** Please follow the discussion with reviewer `#zrrF`.
>
> ---
>
> [1] Ricky T. Q. Chen, Brandon Amos, Maximilian Nickel, "Learning Neural Event Functions for Ordinary Differential Equations", ICLR 2021.

---

### Official Review · Reviewer_Q3GY · 2021-11-05

**Correctness:** 4
**Technical Novelty And Significance:** 3
**Empirical Novelty And Significance:** 3
**Recommendation:** 6
**Confidence:** 3

**Main Review:**

+ new domain using Neural ODE
+ well written
+ well-thought domain-specific implementation tricks to adopt Neural ODE
+ evaluated with multiple drawing datasets
- limited domain experiment
- incomplete ablation study
- heavy dependence on the original Neural ODE work

The paper presented a list of implementation tricks to improve the performance of their Neural-ODE-enabled sequential model. The ideas of using the Perlin Noise and periodic activation functions are up-to-date and show their effectiveness to a certain degree.

Also I greatly appreciate the authors providing the details for working with the Multi-stroke format data, not just with Full-sequence format.

The paper is very well written, but it was difficult to assess the level of novelty. The authors claim that this is a new continuous-time sequential model, but they only tested their model with drawing datasets, that seems to me that there is a little bit of overstatement. This paper could be improved if the authors provide experimental results in other sequential domains, such as speech. Also, adding to that, it would be most appreciative if the authors could add a list of main contributions when this is compared against Neural ODE, as many ideas are dependent on this previous paper and it made me difficult to easily evaluate the level of contributions.

Finally, it could have been better if other new methods that work with sequential data, such as transformer-enabled models, are added to the baseline. I found it very odd that the authors did not include any of those methods for comparison.

**Summary Of The Paper:**

This paper presents a novel approach to represent chirographic drawing data with Neural ODE and demonstrates its effectiveness by comparing with the standard baseline Seq2Seq model.

**Summary Of The Review:**

Overall, the paper is very well-written and shows fascinating results with continuous-time seq2seq model.
However there are still some missing components in the paper, and more clarification needed to distinguish this method from the Neural ODE work.

---

> ### Author Response · Authors · 2021-11-22
> **Official response to reviewer Q3GY**
>
> We thank you for recognizing the novelties (new domain, design tricks) of the paper and appreciating the clarity of writing.
>
> **Clarity on core contributions:** On your request to a list of points depicting the main contributions, reviewer `#zrrF` already did so quite elegantly. We would like to clarify it once again with few more bits:
>
> 1. Our paper proposes to model chirographic data entirely as a generic function of time $\mathbf{s}(t)$ which can be trained with already available discrete points. Earlier works have tried doing the same using specific functional forms like Discrete SVG commands [1], parametric curves [2] OR separate "Stroke + Relational" model [3].
> 2. We do the above by introducing (for the first time) an autoencoder structure, consisting of a Neural CDE based encoder and Neural ODE based decoder. Please note that **we do not claim to alter** the existing theoretical works (including training mechanisms) done in ODE/CDE literature in any way. However, we did introduce some design choices (2nd order ODE, multi-stroke representation) and implementation tricks (periodic activation, Perlin noise augmentation) geared towards the specific domain of application.
> 3. We are also the first to shed light onto some useful properties that stem out of "*inherent continuity*" and "*inductive bias*" exerted by ODE systems. These properties are consequences of both encoder and decoder being deterministic, which is just not possible with methods that have autoregressive components [2,3].
> 4. We also noticed the property of "*abstraction*" stemming from our own design choice (i.e. periodic activation). Works on chirographic abstraction are very limited and only work with selection of stroke subsets which is much more "inelegant" than our method.
>
> **More generic method and overstatements:** We agree with you that we made statements like "First continuous-time Seq2Seq model" and focused on one specific domain (chirographic data). But it's also true that our Seq2Seq architecture is indeed quite generic. In fact, the description of the framework up to `section 4.3` has almost no assumption about the type of data and still qualifies to be called a Seq2Seq model. However, we purposefully wanted to keep the paper geared towards chirographic data as it serves as a "benchmark for continuous time data" as reviewer `#S7jh` correctly mentioned. Applying this architecture to more generic continuous-time domains like audio (your suggestion), video are surely possible but may require slightly different design choices and more rigorous experiments which would really require writing a completely different paper to execute properly. We’ll be happy to pursue this as a future work. We will make this clear in the final version.
>
> **Transformer-based models:** The focus of the paper is to explore some interesting and emergent properties (data-efficiency, inherent continuity etc.) of continuous-time models. With regards to these specific properties, RNNs and Transformers have quite similar behavior (both autoregressive) and hence we decided to keep only RNNs as a baseline.
>
> ---
>
> [1] Alexandre Carlier, Martin Danelljan, Alexandre Alahi, and Radu Timofte. Deepsvg: A hierarchical generative network for vector graphics animation.NeurIPS, 2020.
>
> [2] Ayan Das,  Yongxin Yang,  Timothy Hospedales,  Tao Xiang,  and Yi-Zhe Song. "BezierSketch:  A generative model for scalable vector sketches". ECCV, 2020.
>
> [3] Emre Aksan, Thomas Deselaers, Andrea Tagliasacchi, and Otmar Hilliges. "CoSE: Compositional stroke embeddings". NeurIPS, 2020.

---

### Decision · Program_Chairs · 2022-01-20

**Decision:**

Accept (Poster)

**Comment:**

The SketchODE submission is a continuously-valued model for chirographic drawing data such as handwritten digits or sketches. It relies on variational sequence-to-sequence model where the latent code z is a global encoding of the drawing dynamical, and contains a neural controlled differential equation encoder to encode a discrete 2D drawing sequence s, and an augmented neural ODE decoder (conditional on the latent code z) to model both the first-order dynamics both of the drawing velocity and of the pen state (effectively modelling second order dynamics on the pen position). The model enables to sample sketches by sampling latent codes, as well as to interpolate between two latent codes, and is evaluated on VectorMNIST (a new task), QuickDraw sketches, and DiDi schematics, where it is compared to discrete RNN-based Seq2Seq and two more recent baselines.

Reviewers praised the idea of using continuously-valued Neural ODEs for drawing, compelling properties of the model for conditional generation or interpolation, the new VectorMNIST dataset, and the writing. Reviewers had some concerns: overstating the novelty and contribution to general continuous seq2seq given that the evaluation was done only on chirographic drawing tasks (Q3GY, zrrF), some experimental details such as missing ablations, examples from QuickDraw or Didi, or comparisons with transformers (Q3GY, zrrF), clarifications on computational complexity (zrrF, S7jh), situating the work with respect to applications of Neural ODEs to physics (zrrF); most of these concerns were addressed in the rebuttal. Reviewer KvGm had the most concerns about the experimental section, but has increased their score after the discussion with the authors.

There was no discussion among the reviewers, only between the authors and reviewers zrrF and KvGm. After the authors' rebuttal, the scores became 8. 8, 6 and 5, and thus I believe that the paper meets the conference acceptance bar.